# Cognitive Study Strategies and Motivational Orientations among University Students: A Latent Profile Analysis

Conny De Vincenzo [1,*] and Matteo Carpi [2,*]

1   Department of Education Science, Roma Tre University, 00185 Rome, Italy
2   Department of Human Neurosciences, Sapienza University of Rome, 00185 Rome, Italy
*   Correspondence: conny.devincenzo@uniroma3.it (C.D.V.); matteo.carpi@uniroma1.it (M.C.)

**Abstract:** Cognitive study strategies and motivational orientations play a crucial role in promoting successful learning and academic achievement. This study aims to explore the presence of different cognitive–motivational profiles among Italian university students using latent profile analysis. A sample of 476 university students (mean age: 26.5 ± 9.6 years; 71.6% female) participated in a cross-sectional online survey, including the Self-Regulated Knowledge Scale-University, the Academic Motivation Scale, the Students' Self-Efficacy Scale, the Outcome Questionnaire-45, and a brief measure of dropout intention. Latent profile analysis identified three distinct configurations of self-regulated learning strategies and motivational orientations: "autonomous-motivated deep learners" (AUT-Learn; 60.5%), "externally-motivated balanced strategists" (EXT-Bal; 36.8%), and "externally-motivated task performers" (EXT-Task; 6.8%). The three profiles exhibit significant differences in self-efficacy, dropout intention, grade point average (GPA), and psychological wellbeing. The AUT-Learn profile showed higher self-efficacy than EXT-Bal and EXT-Task, and higher GPA than EXT-Task. Additionally, AUT-Learn participants reported lower dropout intention and higher psychological wellbeing compared to EXT-Bal and EXT-Task, while EXT-Bal showed lower dropout intention and better wellbeing than EXT-Task. The predictive validity and practical implications of this classification warrant further investigation in dedicated longitudinal studies.

**Keywords:** self-regulated learning; study strategies; motivation; university students; university dropout; latent profile analysis



## 1. Introduction

The ability to self-regulate one's learning is associated with several positive outcomes, including better academic performance and lower dropout rates [1–4]. Emotional and motivational components, alongside cognitive aspects, play a strategic role in promoting study success [5].

Self-regulated learners take an active and constructive role in their own learning process. They are able to monitor, control, and regulate cognitive, motivational, behavioral, and contextual aspects to align them with their learning objectives. These learners can identify the most effective learning strategies and adapt them based on their reflections and the external environment [6,7]. Additionally, self-regulated learners demonstrate awareness of effective cognitive and motivational strategies, continually evaluating their learning process and adjusting their strategies as necessary [4,8].

Cognitive regulation refers to the strategies used to learn and perform a given task, such as repetition, rehearsal, organization, critical thinking, and monitoring of learned material. Motivational regulation, on the other hand, involves managing different types of motivation to learn [7,9]. One of the leading meta-analyses in the field [9] showed that, with specific reference to cognitive strategies, critical thinking, and processing strategies (indicative of deep learning) were positively associated with a higher grade point average. Earlier work by Robbins and colleagues [10] also demonstrated that the most important

predictors of persistence were the presence of academic goals, self-efficacy, and a set of academic competencies. However, in terms of grade point average, the main predictors were self-efficacy and achievement motivation.

Regarding academic motivation, Ryan and Deci's self-determination theory describes the different types of motivational regulation that underlie learning processes [11–13]. Specifically, there are five types of motivational regulation that vary along a continuum from extrinsic to intrinsic regulation. Extrinsic regulation is the classic form of extrinsic motivation, where behavior is controlled by external contingencies, such as obtaining a reward or avoiding a punishment. This type of regulation is considered the most controlled and least autonomous. Moving along the continuum, introjected regulation involves non-self-regulated behavior, where an individual engages in an activity to avoid negative states of mind, such as guilt or anxiety, or to enhance self-image. A more autonomous form of extrinsic motivation is identified regulation, where the behavior becomes more self-endorsed and is integrated into the individual's identity. The most autonomous form of extrinsic motivation is integrated regulation, which involves not only identifying with the meaning of the behavior but also integrating these identifications with other aspects of the self. At the end of the continuum is intrinsic motivation, the most autonomous form of regulation, characterized by engaging in an activity for the inherent pleasure and satisfaction derived from it. More autonomous forms of motivation have been shown to promote better academic performance and greater wellbeing [14–16].

Numerous studies have examined the impact of cognitive strategies and study motivation on academic success [3,17–21]. For example, Manganelli et al. analyzed the effect of cognitive strategies and study motivation on academic performance in a sample of university students using structural equation modeling [3]. The results showed the direct contribution of cognitive strategies, particularly critical thinking, to academic performance. They also found an indirect effect of autonomous motivation on performance through critical thinking, and a direct effect of controlled motivation on academic performance. Additionally, a paper by Bernardo et al. demonstrated that the intention to stay at university was predicted by beliefs about the effectiveness of different self-regulation strategies used in studying, as well as by student satisfaction [19].

Notably, most of the studies examining the impact of self-regulation strategies on academic performance have employed a variable-centered approach, which operates at the group level and aims to explain the relationships between a set of variables. In contrast, person-centered approaches, such as latent class or latent profile analysis, aim to identify differentiated subgroups of individuals who share certain characteristics [22].

Such approaches can clarify how students integrate and combine different motivational and learning strategies effectively, contributing to the delineation of a self-regulated learner profile. In the educational context, some studies have used a person-centered approach [23–25], highlighting differences in academic performance and emphasizing the practical implications of differentiated profiles.

This study aims to explore the presence of different cognitive–motivational profiles among Italian university students using latent profile analysis and to investigate the differences between the profiles with respect to academic performance, dropout intention, self-efficacy, and psychological wellbeing.

## 2. Materials and Methods

### 2.1. Participants and Procedure

Participants were 476 university students (mean age: $26.5 \pm 9.6$ years; 341 women and 127 men) who took part in a survey aimed at exploring the determinant of dropout intention and academic success. The study design was cross-sectional, and data were collected via an anonymous online survey delivered to students through the Limesurvey platform from February 2022 to July 2022. Participation in the study was voluntary, and participants provided online informed consent before completing the survey. All research procedures were conducted in accordance with the Ethical Code of the Italian Association

of Psychology (AIP) and the Declaration of Helsinki, as revised in 2013. Since all the instruments administered were answered anonymously, it was not considered necessary to obtain ethical approval for this study.

*2.2. Measures*

Sociodemographic data and information about academic career (including GPA) were obtained through ad hoc questions, while cognitive strategies were assessed using the Self-Regulated Knowledge Scale-University (SRKS-U) [26], academic motivation was investigated with Academic Motivation Scale (AMS) [27], dropout intention with an adapted version of Hardre and Reeve's scale [28,29], academic self-efficacy with a modified version of the Italian Perceived School Self-efficacy Scale [30,31], and psychological wellbeing with the Outcome Questionnaire-45 [32,33]. GPA was self-reported by participants and recorded as a continuous variable ranging from 18 to 30, reflecting the standard Italian university grading system.

Internal consistency was estimated through McDonald's omega coefficient ($\omega$) to evaluate the scale reliability [34].

### 2.2.1. Self-Regulated Knowledge Scale-University

The Self-Regulated Knowledge Scale-University (SRKS-U) [26] is a self-report questionnaire that assesses the main cognitive strategies used during study and is based on Pintrich's self-regulation model of learning [7,35]. The SRKS-U consists of 15 items, where students indicate how often they use specific cognitive strategies during their studies on a 5-point Likert scale (from "never" to "always or almost always"). The scale comprises five subscales, each measuring a specific cognitive strategy: knowledge extraction, knowledge networking, knowledge practice, knowledge critique, and knowledge monitoring.

The knowledge extraction subscale assesses the organization of learned material through diagrams and summaries (e.g., "I make summaries of the most important things"). The knowledge networking subscale measures the processing and linking of learning material (e.g., "I look for similarities or differences between what I am studying and what I already know"). The knowledge practice subscale evaluates the frequency of rehearsal (e.g., "I repeat to myself several times what is important to learn"). Finally, the knowledge critique assesses the use of critical thinking (e.g., "I try to make up my own mind about the things I am studying"), while knowledge monitoring assesses learning monitoring (e.g., "I check that I have understood what I have read correctly"). The total score for each subscale was obtained by summing the item scores. McDonald's omega coefficient of the subscales in the sample ranged from 0.71 to 0.82.

### 2.2.2. Academic Motivation Scale

Study motivation was assessed using the Italian version of the Academic Motivation Scale [27,36,37], which is based on Ryan and Deci's self-determination theory. The AMS consists of 20 items that explore students' reasons for choosing their course of study, using an 11-point response scale ranging from 0 ("not at all true") to 10 ("completely true"). Autonomous motivation is represented by the subscales "identified regulation" (e.g., "Because it is important for what I have decided to do in the future") and "intrinsic motivation" (e.g., "Because I like the course I have chosen"). Controlled motivation is represented by the subscales "external motivation" (e.g., "Because someone else is pushing me to do it") and "introjected motivation" (e.g., "Because I would feel proud of myself if I completed this course"). The Italian version of the scale validation study also identified a fifth subscale, "amotivation", which accounts for the lack of academic motivation without regulating effort [27]. Consistent with the purpose of examining motivational regulation along the continuum from intrinsic to extrinsic motivation, we chose not to include the "amotivation" subscale in this study and considered only the four "identified regulation", "intrinsic motivation", "external motivation", and "introjected motivation" subscales. Scores for each

motivational orientation were calculated by summing the item scores for the subscales. McDonald's omegas of the subscales ranged from 0.80 to 0.92 in this study.

### 2.2.3. Dropout Intention Scale

Students' intentions to drop out of university were evaluated using a modified version of the Hardre and Reeve's scale [28] adapted to the Italian university context [29]. The questionnaire consists of four items, asking students to indicate on a five-point response scale (from "never" to "always or almost always") how often they have thought about dropping out and doing something else, felt insecure about continuing their studies, and considered leaving university. The total score was the mean of the item scores. The instrument showed good reliability in this study ($\omega = 0.92$).

### 2.2.4. Academic Self-Efficacy

A 9-item version of the school self-efficacy scale developed by Pastorelli and Picconi [30] adapted for university students [31] was used to evaluate students' self-efficacy in studying and their beliefs regarding their ability to organize their studies, regulate their motivation while studying, stay focused, and commit. Students respond to items on a 5-point scale (ranging from "not at all able" to "completely able"), indicating their perceived ability to handle specific situations that may arise during their course of study. The mean of the 9 item scores was taken as the total self-efficacy score. In this study, the academic self-efficacy scale showed good internal consistency ($\omega = 0.88$).

Since the dimensional structure of this adapted scale had not been previously tested, an exploratory factor analysis with oblimin rotation was performed, and the number of factors to be extracted was determined by parallel analysis [38]. A one-factor solution was identified, with factor loadings greater than 0.53 for all items.

### 2.2.5. Outcome Questionnaire-45

The Outcome Questionnaire-45 (OQ-45) [32,33,39,40] is a 45-item self-report questionnaire that provides information about an individual's general functioning and wellbeing. It examines some common problems and symptoms associated with mental health problems. Participants rate how often they experience certain common situations on a five-point frequency scale (from "never" to "almost always"). Total scores range from 0 to 180, with higher scores indicating more problematic general functioning. In addition to the total score, the OQ-45 provides scores for three subscales: (a) Symptomatic distress, which examines the presence of common psychological symptoms such as anxiety and depression; (b) interpersonal relations, which assesses problems in interpersonal relationships; and (c) social role, which detects difficulties in various social contexts, such as school, university, and work. The OQ-45 has been used in research to evaluate the effectiveness of clinical interventions [41].

Given the limited consistency of specific dimensions in previous studies [33], we chose to consider only the OQ-45 total score as a measure of psychological wellbeing in this study. McDonald's omega for the total score was 0.95 in our sample.

### 2.3. Statistical Analyses

Continuous variables were summarized using means and standard deviations, and categorical variables were summarized using counts and percentages. Bivariate associations between variables were assessed using Pearson's correlation coefficients.

Latent profile analysis (LPA) was used to evaluate alternative classification solutions for subgrouping participants based on self-reported learning strategies and motivational orientations.

LPA is a statistical approach belonging to finite mixture models, a special type of latent variable models that assume that the distribution of a set of variables can be synthesized as a mixture of a finite number of latent variables distributions, i.e., latent classes or profiles [42,43]. Thus, LPA allows for the identification of heterogeneous subgroups of

individuals (profiles) based on their scores on predetermined indicator variables and assigns each individual to a profile with a given probability.

In this study, the SRKS-U and AMS subscales were considered as indicators. For ease of interpretation and visualization, LPA was performed using z-scores with a mean of 0 and a standard deviation of 1 for each variable, and models with 1 to 6 profiles were estimated to evaluate a wide range of solutions comparatively. A combination of conventional fit indices and theoretical significance was used to identify the best fitting model. According to published guidelines [44], the following indices and model parameters were considered: Akaike's information criterion (AIC) and its consistent version (CAIC), Bayesian information criterion (BIC) and its sample size adjusted version (SABIC), bootstrapped likelihood ratio test (BLRT), entropy (i.e., the degree to which the model identifies well-separated profiles [45]), and minimum and maximum posterior classification probabilities. Lower information criteria values, significant BLRT, higher entropy, and higher posterior probabilities were considered indicative of a better model fit [44,46].

Differences between profiles on indicator (i.e., SRKS-U and AMS subscales) and outcome (GPA, dropout intention, self-efficacy, and psychological wellbeing) variables were assessed using one-way analyses of variance (ANOVA). Eta-squared ($\eta^2$) effect sizes were calculated for each comparison, and Games–Howell post hoc tests were performed for pairwise comparisons to control for possible unequal variances. The chi-square ($\chi^2$) test was used to evaluate differences in gender proportions between profiles.

All analyses were performed using jamovi v. 2.5 [47] and R v. 4.4.1 [48]. The tidyLPA R package [49] was used for LPA, assuming equal variances and covariances fixed at 0 for the indicator variables in each profile. *p*-values less than 0.05 were considered statistically significant.

## 3. Results

### 3.1. Participants' Characteristics

Participants' demographic characteristics and information about their academic careers are shown in Table 1. The majority of the sample was female (341; 71.6%) and full-time students (266; 55.9%). Most students were enrolled in undergraduate courses (411; 86.3%), i.e., three-year undergraduate degrees or five-year program in educational science, while only 13.7% (65) of participants were enrolled in postgraduate programs (i.e., two-year master's degree programs). Mean scores for the administered questionnaire for the whole sample are reported in Table 2, and bivariate correlations between SRKS-U and AMS subscales are reported in Table A1 (Appendix A).

**Table 1.** Demographic and academic characteristics of the study sample (*N* = 476).

| Variable | N (%) | Mean (SD) |
|---|---|---|
| Sex | | |
| Women | 341 (71.6) | |
| Men | 127 (26.7) | |
| Prefer not to report | 8 (1.7) | |
| Age | | 26.5 (9.6) |
| Occupational situation | | |
| Full-time student | 266 (55.9) | |
| Part-time job | 121 (25.4) | |
| Full-time job | 89 (18.7) | |
| Study course | | |
| Undergraduate | 411 (86.3) | |
| Postgraduate | 65 (13.7) | |
| Grade point average [a] | | 26.3 (2.6) |

[a]: *n* = 55 missing values.

**Table 2.** Means and standard deviations and statistical comparisons of questionnaire scores and grade point average for the total sample and the three profiles identified with latent profile analysis. Possible score ranges are reported in brackets for each questionnaire.

| Variable | Total Sample (N = 476) Mean (SD) | AUT-Learn (n = 288) (1) Mean (SD) | EXT-Bal (n = 156) (2) Mean (SD) | EXT-Task (n = 32) (3) Mean (SD) | F | $\eta^2$ | Sig. Post-Hoc |
|---|---|---|---|---|---|---|---|
| SRKS-U Extraction (3–15) | 11.8 (2.8) | 12.3 (2.7) | 10.9 (2.8) | 11.7 (2.9) | 13.2 *** | 0.05 | 1:2 |
| SRKS-U Networking (3–15) | 11.3 (2.6) | 12.7 (1.7) | 9.10 (2.2) | 8.9 (2.6) | 206.7 *** | 0.47 | 1:2; 1:3 |
| SRKS-U Practice (3–15) | 12.2 (2.2) | 12.9 (1.9) | 11.4 (2.1) | 10.5 (2.6) | 39.7 *** | 0.14 | 1:2; 1:3 |
| SRKS-U Critique (3–15) | 10.6 (2.8) | 12.1 (1.9) | 8.3 (2.0) | 8.1 (2.9) | 215.5 *** | 0.48 | 1:2; 1:3 |
| SRKS-U Monitoring (3–15) | 12.7 (1.9) | 13.6 (1.3) | 11.7 (1.7) | 10.1 (2.3) | 124.8 *** | 0.35 | 1:2; 1:3; 2:3 |
| AMS External (0–40) | 4.0 (7.1) | 2.6 (5.6) | 4.3 (6.3) | 15.3 (11.8) | 56.9 *** | 0.19 | 1:2; 1:3; 2:3 |
| AMS Introjected (0–40) | 19.0 (10.5) | 18.2 (10.3) | 20.5 (10.7) | 17.7 (11.1) | 2.7 | 0.01 | – |
| AMS Identified (0–40) | 31.8 (9.5) | 32.9 (9.2) | 31.8 (9.0) | 22.3 (9.0) | 19.4 *** | 0.08 | 1:3; 2:3 |
| AMS Intrinsic (0–40) | 34.5 (6.2) | 37.0 (3.6) | 33.2 (4.6) | 19.1 (6.8) | 269.2 *** | 0.53 | 1:2; 1:3; 2:3 |
| GPA [a] | 26.3 (2.6) | 26.6 (2.5) | 26.0 (2.6) | 25.1 (3.0) | 6.0 ** | 0.03 | 1:3 |
| Self-efficacy (1–5) | 3.3 (0.7) | 3.6 (0.6) | 3.1 (0.5) | 2.8 (0.8) | 49.6 *** | 0.17 | 1:2; 1:3 |
| Dropout (1–5) | 2.1 (1.0) | 1.8 (0.8) | 2.2 (0.9) | 3.3 (1.1) | 43.2 *** | 0.15 | 1:2; 1:3; 2:3 |
| OQ-45 (0–180) | 58.4 (26.1) | 54.8 (25.4) | 61.5 (25.2) | 76.1 (27.9) | 11.7 *** | 0.05 | 1:2; 1:3; 2:3 |

[a]: *n* = 55 missing values for GPA, 32 for AUT-Learn, 17 for EXT-Bal, and 6 for EXT-Task. *Note.* AUT-Learn: autonomously-motivated deep learners; EXT-Bal: externally-motivated balanced strategists; EXT-Task: externally-motivated task performers; SRKS-U: Self-Regulated Knowledge Scale-University; AMS: Academic Motivation Scale; GPA: grade point average; OQ-45: Outcome Questionnaire-45. ** $p < 0.01$; *** $p < 0.001$.

### 3.2. Learning Strategies and Motivational Orientation Profiles

The results of the LPA conducted with cognitive strategies and motivational orientations as indicator variables are reported in Table 3. Combining fit indices and theoretical coherence, a three-profile solution was selected. Although the decreasing values of the information criteria indices (AIC, CAIC, BIC, SABIC) could indicate a better fit for models with more profiles, the chosen three-profile model showed the highest entropy value along with an optimal balance between the minimum and maximum posterior classification probabilities. Moreover, the three profiles parsimoniously accounted for the expected differences in cognitive–motivational strategies compared to more complex models, with no apparent loss of information.

Based on the respective configurations of learning strategies and motivational orientation scores shown in Figure 1, the three identified profiles were named as follows: autonomously-motivated deep learners (AUT-Learn), comprising the majority of participants (60.5%, *n* = 288) and characterized by a more autonomous and intrinsic motivational attitude and frequent use of all assessed cognitive strategies (especially deeper strategies such as critical thinking and monitoring); externally-motivated balanced strategists (EXT-Bal; 36.8%, *n* = 156), comprising participants with an externally driven academic motivation

and low to average levels of cognitive strategies, with a predominant use of extraction and practice strategies; externally-motivated task performers (EXT-Task), comprising a minority of the sample (6.8%, *n* = 32), characterized by an external motivation with low intrinsic motivation scores and low cognitive strategies scores, except for the more superficial extraction strategy (i.e., use of diagrams and summaries to support learning).

**Table 3.** Fit indices obtained for the models examined with latent profile analysis.

| Model | AIC | CAIC | BIC | SABIC | BLRT *p*-Value | Entropy | Prob min–max |
|---|---|---|---|---|---|---|---|
| 1 profile | 12,184 | 12,277 | 12,259 | 12,202 | – | – | – |
| 2 profiles | 11,693 | 11,837 | 11,809 | 11,721 | 0.01 | 0.789 | 0.90–0.95 |
| 3 profiles | 11,530 | 11,726 | 11,688 | 11,568 | 0.01 | 0.815 | 0.87–0.94 |
| 4 profiles | 11,460 | 11,708 | 11,660 | 11,508 | 0.01 | 0.748 | 0.78–0.91 |
| 5 profiles | 11,169 | 11,468 | 11,410 | 11,226 | 0.01 | 0.804 | 0.79–0.96 |
| 6 profiles | 11,103 | 11,455 | 11,387 | 11,171 | 0.01 | 0.812 | 0.80–0.95 |

*Note.* AIC: Akaike information criterion; BIC: Bayesian information criterion; SABIC: sample size-adjusted Bayesian information criterion; BLRT: bootstrap likelihood ratio test; Prob min–max: minimum and maximum posterior classification probabilities.

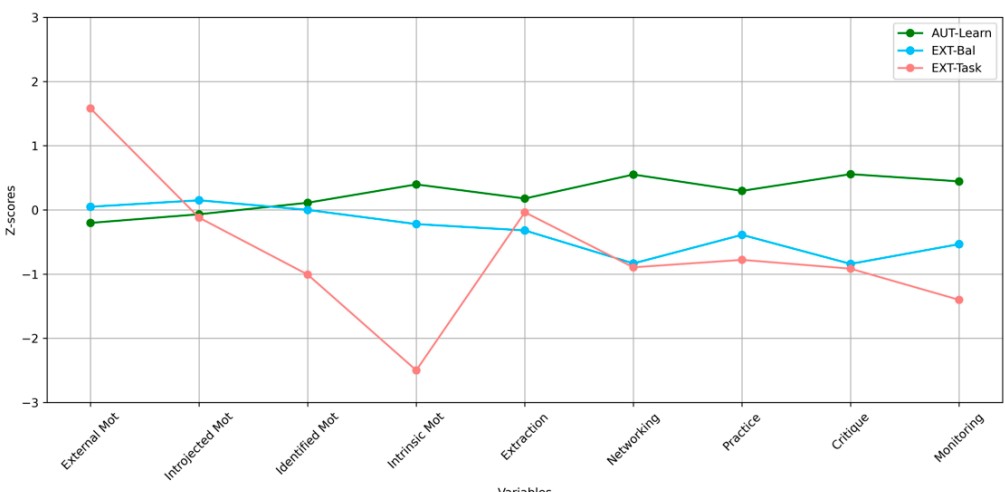

**Figure 1.** Z-scores of motivational orientations assessed with the Academic Motivation Scale and cognitive strategies evaluate with the Self-Regulated Knowledge Scale-University for the three profiles identified with latent profile analysis. Note: AUT-Learn: autonomously-motivated deep learners; EXT-Bal: externally-motivated balanced strategists; EXT-Task: externally-motivated task performers.

The results of the statistical comparisons for the indicator variables between the three groups are shown in Table 2. In particular, AUT-Learn showed higher scores in networking, practice, monitoring, and critique strategies compared to the other two profiles, and a higher extraction score compared to EXT-Bal. On the other hand, a significant difference in monitoring was observed between EXT-Bal and EXT-Task, with the latter showing a lower mean score. Regarding motivational orientations, as expected, all profiles differed in intrinsic and external motivations, whereas AUT-Learn was characterized by a higher identified motivation score compared to both EXT-Bal and EXT-Task. No differences were found between the profiles in introjected motivation.

### 3.3. Differences in Academic and Psychological Outcomes between Profiles

No differences were observed in the proportions of female and male students in the three profiles ($\chi^2$ = 1.9, degrees of freedom = 2, *p* = 0.38).

The lower part of Table 2 illustrates the differences in GPA, perceived academic self-efficacy, dropout intention, and psychological wellbeing between the three profiles. Significant differences were found for all of the outcome variables considered. Specifically, the three groups differed in terms of dropout intention and OQ-45 score, with AUT-Learn showing lower dropout intention and better wellbeing than EXT-Bal and EXT-Task, and EXT-Bal showing a better condition (i.e., higher wellbeing and lower dropout intention) than EXT-Task. In addition, students in the AUT-Learn group also reported higher academic self-efficacy than their peers in the other two groups and a significantly higher GPA than EXT-Task participants.

## 4. Discussion

The purpose of this study was to examine the presence of differential profiles in cognitive strategies and motivational orientations among a sample of college students using latent profile analysis. The results revealed three distinct groups that differed in terms of psychological and academic outcomes. Specifically, the group with the best cognitive motivational profile, labeled autonomously-motivated deep learners (AUT-Learn), was characterized by an autonomous motivational orientation and a frequent use of deep cognitive strategies such as critical thinking and monitoring. This configuration was associated with lower dropout intentions, better perceived wellbeing, higher GPA, and higher self-efficacy. On the other hand, the group of externally-motivated balanced strategists (EXT-Bal) showed a cognitive–motivational profile characterized by a mild external motivational orientation and a medium level of cognitive strategies, with a predominance of surface strategies such as extraction and practice. Notably, this group did not differ from the others in GPA but scored in the middle in terms of dropout intention and wellbeing. Finally, the least self-regulated profile, i.e., externally-motivated task performers (EXT-Task), was characterized by strong external motivation and low reliance on functional cognitive strategies in studying, with a prevalence of superficial cognitive strategies. Consistently, this cognitive–motivational profile also showed higher dropout intentions, lower wellbeing, lower GPA, and lower self-efficacy compared to both EXT-Bal and AUT-Learn.

Taken together, the three profiles appear to lie along a cognitive–motivational continuum, ranging from those with the most self-regulated profile (AUT-Learn), through those with a primarily external-oriented profile and moderate reliance on cognitive strategies (EXT-Bal), to those with the least self-regulated profile (EXT-Task).

Overall, these findings are consistent with well-established educational research showing a strong link between self-regulation and motivational dispositions [3,16]. They also highlight the potential of person-centered approaches in identifying individual, specific configurations that may be related to differential behaviors, attitudes, and academic outcomes.

To our knowledge, only one previous study by Broadbent and Fuller-Tyszkiewicz [24] used LPA to examine cognitive–motivational strategies, taking into account a wide range of indicator variables, including motivational and self-regulated learning strategies, as measured by the instrument developed by Pintrich [50], grade goals, study automaticity [51], and intention to study. The authors identified five distinct profiles: "restrained regulators" and "minimal regulators" showed intermediate and low scores, respectively, in all domains assessed. On the other hand, both "capable collaborators" and "self-reliant capable regulators" obtained high scores in cognitive–motivational strategies, with the former showing high test anxiety and the latter characterized by low test anxiety and low scores in learning strategies involving peer collaboration. Finally, "super-regulators" had the highest scores in all cognitive and motivational strategies. In terms of academic performance, "super-regulators" and "self-reliant capable regulators" reported higher GPA, while "minimal regulators" reported lower grades. In addition, despite gender differences previously being reported in academic motivation and cognitive strategies [52–54], we did not find differences in the proportions of male and female students between profiles.

Consistent with our findings, the profiles characterized by high levels of self-regulation, i.e., "capable collaborators" and "super-regulators", showed greater reliance on critical

thinking and more intrinsic motivation, whereas those with lower levels of self-regulation ("restrained regulators" and "minimal regulators") had a more extrinsic motivational orientation along with a predominant use of the superficial learning strategy of rehearsal and reduced metacognitive monitoring. This correspondence appears to support a cognitive–motivational continuum, with deep cognitive strategies mostly expressed at the intrinsic pole of motivation, which fosters the expression of critical thinking and self-monitoring, leading to better academic performance.

From an applied perspective, our findings could inform guidance programs and tailored preventive and ongoing interventions to promote academic success. Specific evidence-based strategies and programs [55–57] can be adapted for the identified subgroups based on their cognitive–motivational profiles and their particular strengths and difficulties. For example, students with a configuration similar to the EXT-Bal and EXT-Task profiles could be encouraged to develop more intrinsic motivational attitudes and to strengthen deep learning strategies through progressive feedback. In this sense, the identification of valued, intrinsic goals could be an asset for building more diverse learning approaches and strengthening self-efficacy, especially for EXT-Task students, who could be helped to construct a more integrated learning plan beyond task performance.

However, despite being one of the first investigations in this field, this study has several limitations. First, participants were recruited from a convenience sample of moderate size, which is not representative of Italian university students. Furthermore, given that participants took part in the survey voluntarily and that it was not possible to estimate the total sample reached, selection bias cannot be ruled out. In fact, it is possible that students more interested in the research topic and more involved in the academic context responded more frequently, limiting the generalizability of our results. Such a mechanism could also explain the low variability in GPA and dropout intention and the overall medium to high scores observed in cognitive strategies, as well as the lack of a distinctive profile characterized by difficulties in learning strategies and impaired academic outcomes. In addition, it was not possible to access academic records, and only self-reported GPA was collected as a measure of academic performance, which may have led to biased estimates of performance. Second, self-regulated learning encompasses numerous cognitive, emotional, and motivational components of learning [7], and our research may have only partially captured the complexity of this construct, omitting important dimensions such as the ability to set learning goals and the ability to manage time [20,58]. Considering that significant correlations were found between the indicators used to identify latent profiles in this study, which may affect the robustness of the LPA results, the investigation of alternative clustering solutions based on broader sets of independent variables is warranted. Finally, the cross-sectional design of the study prevents causal interpretation of the results. In particular, the reported associations between profile membership and academic and psychological outcomes cannot be considered consequential, and our hypotheses about their meanings should be considered speculative. Further research should be conducted with repeated assessments over time (e.g., using latent transition analysis [59]) to examine the consistency and trajectories of the identified profiles over time and their predictive validity with respect to other outcomes.

## 5. Conclusions

In conclusion, this study identified three distinct profiles based on cognitive strategies and motivational orientations in a sample of university students. The three subgroups showed differences in academic outcomes, such as GPA and dropout intention, and in psychological outcomes, with the more intrinsically motivated profile being more self-regulated and reporting higher grades, lower dropout intention, higher self-efficacy, and lower psychological distress. Identifying homogeneous student profiles may be useful in designing targeted and specific interventions to promote academic success. Future studies should further investigate the validity and stability of these subgroups using longitudinal research designs.

**Author Contributions:** Conceptualization, C.D.V.; methodology, C.D.V. and M.C.; formal analysis, C.D.V. and M.C.; investigation, C.D.V.; data curation, C.D.V. and M.C.; writing—original draft preparation, C.D.V. and M.C.; writing—review and editing, C.D.V. and M.C.; project administration, C.D.V. All authors have read and agreed to the published version of the manuscript.

**Funding:** This research received no external funding.

**Institutional Review Board Statement:** The research was conducted in accordance with the Ethical Code of the Italian Association of Psychology (AIP) and the Declaration of Helsinki as revised in 2013. Since data collection was completely anonymous, it was not considered necessary to obtain ethical approval for this study.

**Informed Consent Statement:** Informed consent was obtained from all subjects involved in the study.

**Data Availability Statement:** The data that support the results of this study are available from the corresponding authors upon reasonable request.

**Acknowledgments:** The authors would like to thank all the students who took part in the study.

**Conflicts of Interest:** The authors declare no conflicts of interest.

**Appendix A**

**Table A1.** Pearson's correlation coefficients between the Academic Motivation Scale (AMS) and the Self-Regulated Knowledge Scale-University (SRKS-U) subscales.

| | 1. | 2. | 3. | 4. | 5. | 6. | 7. | 8. | 9. |
|---|---|---|---|---|---|---|---|---|---|
| 1. SRKS-U Extraction | — | | | | | | | | |
| 2. SRKS-U Networking | 0.20 *** | — | | | | | | | |
| 3. SRKS-U Practice | 0.23 *** | 0.24 *** | — | | | | | | |
| 4. SRKS-U Critique | 0.15 *** | 0.68 *** | 0.16 *** | — | | | | | |
| 5. SRKS-U Monitoring | 0.19 *** | 0.44 *** | 0.58 *** | 0.42 *** | — | | | | |
| 6. AMS External | 0.01 | −0.17 *** | −0.07 | −0.13 ** | −0.16 *** | — | | | |
| 7. AMS Introjected | 0.05 | −0.04 | 0.04 | −0.10 * | 0.00 | 0.31 *** | — | | |
| 8. AMS Identified | 0.12 ** | 0.10 * | 0.13 ** | 0.05 | 0.11 * | −0.22 *** | 0.12 ** | — | |
| 9. AMS Intrinsic | 0.03 | 0.36 *** | 0.21 *** | 0.41 *** | 0.36 *** | −0.39 *** | 0.06 | 0.33 | — |

*Note*. SRKS-U: Self-Regulated Knowledge Scale-University; AMS: Academic Motivation Scale. * $p < 0.05$; ** $p < 0.01$; *** $p < 0.001$.

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
