# Peer review of "Cognitive Study Strategies and Motivational Orientations among University Students: A Latent Profile Analysis"

_education, doi:10.3390/educsci14070792_

Round 1

Reviewer 1 Report

Comments and Suggestions for Authors

Review Cognitive Study Strategies and Motivational Orientations 2 among University Students: A Latent Profile Analysis

Thank you for the opportunity to review this MS focusing on strategies, motivation, self-efficacy, drop out and their relationships. Substantial effort has gone into the research and while there is some potential, that potential is diminished by a number of factors.

Let me elaborate:

·        The literature substantiating the research is very limited in scope and criticality. There is no explicit rationale for why motivation and strategies are predictors of self-efficacy and drop out and GPA. Why aren’t motivation and strategies, through self-efficacy, seen as predictors of drop-out and GPA?  Much more needs to be done to establish a strong rationale for the study and the relationships and directionality among variables.  

·        GPA is mentioned in the Abstract, Table 2 and text but there is no mention of how this was actually measured. (Very small SD?)

·        No mention is made of the factorial validity of the SRKS-U, the AMS and the Self-efficacy measures. Without such data, any clustering of individuals is problematic. Does the SRKS-U have five subscales as suggested? For the AMS it seems that the researchers have been selective. References 34 and 35 indicate six and seven sub-scales respectively and the Italian reference [27] illustrates five factors from CFA. Why have three external oriented styles and one intrinsic style been employed here. What factorial evidence is presented to demonstrate satisfactory validity with the four dimensions chosen?  The Self-efficacy measure seems to be developed from a 14-item scale designed for musicians. How were the nine items determined here? Is it a one-factor measure?

·        The Outcome Questionnaire (OQ-45) also suffers from validity questions. Is it a coherent three factor measure?

·        It is stated that the Drop-out Intention scale was adapted from [28] where three items were employed. Four are used here? Factorial support?

·        Providing latent profile analyses without detailing the correlations among variables is problematic. 

Author Response

We thank the reviewer for their time and their thorough observations that we find helpful in ameliorating our work. Below we report our responses to the reviewer's comments (reviewer's comments are reported in bold). Changes made in the revised manuscript were highlighted in yellow.

The literature substantiating the research is very limited in scope and criticality. There is no explicit rationale for why motivation and strategies are predictors of self-efficacy and drop out and GPA. Why aren’t motivation and strategies, through self-efficacy, seen as predictors of drop-out and GPA?  Much more needs to be done to establish a strong rationale for the study and the relationships and directionality among variables.

We appreciate the reviewer's comment as an opportunity to further reflect and elaborate on the background and scope of our study. Indeed, the literature referenced in the Introduction (e.g., references [3] and [17-21]) largely discusses the rationale for the associations between self-regulation, metacognition, motivational dispositions, and academic outcomes in terms of performance and dropout risk. We have added a missing recall to reference number [3] (Manganelli et al., 2019) in line 64 to make it clear that we are reporting the results of this study. Overall, these previous papers mostly explore linear hypotheses about the causal mechanisms linking these variables, and also highlight possible mediating factors. As mentioned by the reviewer and explored in Nemtcan et al. (2020, reference [20]), from this perspective, self-efficacy (as well as other individual variables such as student integration) could be considered as a mediator in the relationship between cognitive-motivational strategies and dropout intentions. Against this background, our study sought to adopt a distinct approach, as stated in lines 73-77, with the aim of classifying individual cognitive-motivational profiles through finite mixture models. Within this framework, we did not speculate on prediction and causality. The differences between the profiles in terms of dropout intention, GPA, psychological well-being and self-efficacy were examined in order to characterise the clusters using these "outcomes" as external indicators. For this reason, we have only tentatively discussed the hypothesis that a given cognitive-motivational configuration might lead to better or worse academic and psychological conditions. To further reinforce this critical point, in line with the reviewer's suggestion, we have added a dedicated section in the last part of the Discussion dealing with the limitations of our study, with a cautionary statement about the causal interpretation of our cross-sectional, exploratory results. We also mentioned the possibility of further exploring the predictive role of the identified profiles in a longitudinal fashion using latent transition analysis.

GPA is mentioned in the Abstract, Table 2 and text but there is no mention of how this was actually measured. (Very small SD?)

We apologize for this oversight. In the revised manuscript, we have included a description of how GPA was measured. Specifically, we clarified that GPA was self-reported by the students and collected as a continuous variable on a scale from 18 to 30, which is standard in the Italian university system. Given this range of variability, the standard deviation obtained does not necessarily reflect an unusually narrow interval. However, given the considerations reported in the Discussion about a possible selection bias leading to the inclusion of the most motivated and self-regulated students, we added a sentence detailing that the low variability in both dropout intention and GPA could also be explained by this mechanism.

No mention is made of the factorial validity of the SRKS-U, the AMS and the Self-efficacy measures. Without such data, any clustering of individuals is problematic. Does the SRKS-U have five subscales as suggested? For the AMS it seems that the researchers have been selective. References 34 and 35 indicate six and seven sub-scales respectively and the Italian reference [27] illustrates five factors from CFA. Why have three external oriented styles and one intrinsic style been employed here. What factorial evidence is presented to demonstrate satisfactory validity with the four dimensions chosen?  The Self-efficacy measure seems to be developed from a 14-item scale designed for musicians. How were the nine items determined here? Is it a one-factor measure?

We respond to this relevant observation with three separate points for the three measures used:

- Regarding the SRKS-U, we refer to the validation study of the Italian version of the scale (i.e., see [26], Manganelli et al., 2015). In this study, the psychometric properties of the SRKS-U were investigated on a large sample of 2209 Italian university students, and confirmatory factor analysis showed a good fit for a five-factor model compared to a single-factor model and a higher-order factor model (i.e., five first-order factors loading on a general second-order factor). In our research, we used the same five-factor model as Manganelli et al.

- Regarding the AMS, our reference was indeed the Italian validation of the scale by Alivernini & Lucidi (2008, reference [27]). As mentioned by the reviewer, five subscales were identified in their study. We deliberately chose to consider only the four subscales corresponding to intrinsic and extrinsic motivational styles, namely the “identified regulation”, “intrinsic motivation”, “external motivation”, and “introjected motivation subscales”. On the other hand, we excluded the “amotivation” subscale, which did not encompass the motivational continuum of regulation that we were interested in, as was the case in the previous structural equation modeling study by Manganelli et al. (reference [3]). We have explained this choice in detail in the Materials and Methods section of the revised manuscript, in subparagraph 2.2.2. introducing the AMS.

- For the measure of academic self-efficacy, an adapted 9-item version of the original 19-item instrument developed by Pastorelli & Picconi (2001) for school students was used. This short form, specifically designed for university students, was previously published by Biasi et al. (2018). We made this clearer in the revised manuscript by adding a sentence providing a detailed presentation of the adapted scale with an appropriate reference to Biasi et al. Given the limited previous use of this instrument, we also performed an exploratory factor analysis to examine its factor structure, identifying a one-factor solution through parallel analysis. We reported this result in section 2.2.4. As an aside, there is no link with another Italian instrument for measuring self-efficacy in the study of music, developed by Benedetto et al. (2015).

The Outcome Questionnaire (OQ-45) also suffers from validity questions. Is it a coherent three factor measure?

We acknowledge that the factorial validity of the three OQ-45 subscales is much debated in the literature. However, for the Italian version of the measure, Lo Coco et al. (2008, reference [33]) supported a two-level solution with a general factor corresponding to the total score. Acknowledging that several competing solutions for specific factors have been proposed, we chose to consider only the total score as a proxy for psychological well-being in our analyses. We pointed it this choice explicitly in the revised text.

It is stated that the Drop-out Intention scale was adapted from [28] where three items were employed. Four are used here? Factorial support?

We thank the reviewer for pointing out this oversight. The four-item adaptation to the Italian university context, based on the Hardre and Reeve scale, had been presented in another paper by Fagioli (2019), which showed a unidimensional factorial solution for the questionnaire. We made this explicit in the revised manuscript by adding a reference to this previous study, which tested the psychometric properties of the dropout intention measure.

Providing latent profile analyses without detailing the correlations among variables is problematic. 

We acknowledge that bivariate associations between indicator variables are relevant in interpreting the results of latent profile analysis. We have included the results of the correlation analysis in a separate appendix in the revised manuscript.

Although we found some strong significant correlations among cognitive strategies (i.e., networking and critique, r = 0.68, and practice and monitoring, r = 0.58), we ultimately decided to retain all SKRS-U subscales in the latent profile analysis because of their conceptual and theoretical significance.

Reviewer 2 Report

Comments and Suggestions for Authors

This a relevant case study worth reading. Congratulations!

Some minor issues out of curiosity: 

1) You mention in table 2 occupational situation (full-time / part-time). What sort of occupations? Teaching? or other?

2)In section 4 you mention in the discussion that the group with the best cognitive motivational profile (AUT-Learn) was characterized by an autonomous motivational orientation  and use of critical thinking and monitoring. My question is: Is the gender variable relevant here or not?

What about EXT-Bal and &EXT-Task Groups?

If you did not take gender into account, please explain reasons why to the reader. 

Author Response

We thank the reviewer for their time, their appreciation of our work, and their comments. We have tried to improve our work accordingly. Below are our responses to the reviewer's comments (in bold). Changes made in the revised manuscript are highlighted in yellow.

1)You mention in table 2 occupational situation (full-time / part-time). What sort of occupations? Teaching? or other?

This is certainly an important point. The type of employment may indeed be related to the variables studied. Unfortunately, we only collected categorical data from our participants about their occupational status, focusing on the time devoted to the student role (i.e., full-time student, student with part-time job, student with full-time job), without any information about the specific occupation. We will certainly consider this aspect in our future work.

2)In section 4 you mention in the discussion that the group with the best cognitive motivational profile (AUT-Learn) was characterized by an autonomous motivational orientation  and use of critical thinking and monitoring. My question is: Is the gender variable relevant here or not?

What about EXT-Bal and &EXT-Task Groups?

If you did not take gender into account, please explain reasons why to the reader.

The reviewer raised a relevant issue. Indeed, gender differences in motivational orientations and learning strategies have been reported previously. However, we did not find a difference in the proportion of female and male students in the profile we identified. As this aspect was not made clear in the first version of the manuscript, we have highlighted it in the revised text and added appropriate references.

Round 2

Reviewer 1 Report

Comments and Suggestions for Authors

Thanks for the opportunity to review this revised MS. The authors have done a quality revision and addressed my initial concerns to my satisfaction. Thanks for the clarifications. I have only two minor suggestions, if thought appropriate. Should there be some comment in limitations regarding: 1)Self-reported GPA as a potentially biased measure (not actual achievement data), and 2)the use of highly correlated variables in the LPA?

As an aside, my translation programme must have failed me when checking some Italian references. Apologies. 

Author Response

Thanks for the opportunity to review this revised MS. The authors have done a quality revision and addressed my initial concerns to my satisfaction. Thanks for the clarifications. I have only two minor suggestions, if thought appropriate. Should there be some comment in limitations regarding: 1)Self-reported GPA as a potentially biased measure (not actual achievement data), and 2)the use of highly correlated variables in the LPA?

We thank the reviewer for their appreciation and useful additional observations. Mentioning these two aspects among the limitations of the study is indeed appropriate, and so we have modified the manuscript to address the reviewer's suggestion. Specifically, we modified the section dealing with the limitations of the study to explicitly mention the potential bias due to the collection of self-reported GPA as a measure of academic achievement and the potential impact of correlated indicators on the results of the latent profile analysis.

As in the previous round of review, changes made in the revised manuscript are highlighted in yellow.